# Language-guided Manipulator Motion Planning with Bounded Task Space

**Thies Oelerich**[1], **Christian Hartl-Nesic**[1], **Andreas Kugi**[1,2]
[1]Automation and Control Institute (ACIN), TU Wien, Vienna, Austria
[2]AIT Austrian Institute of Technology GmbH, Vienna, Austria
{oelerich,hartl,kugi}@acin.tuwien.ac.at

**Abstract:** Language-based robot control is a powerful and versatile method to control a robot manipulator where large language models (LLMs) are used to reason about the environment. However, the generated robot motions by these controllers often lack safety and performance, resulting in jerky movements. In this work, a novel modular framework for zero-shot motion planning for manipulation tasks is developed. The modular components do not require any motion-planning-specific training. An LLM is combined with a vision model to create Python code that interacts with a novel path planner, which creates a piecewise linear reference path with bounds around the path that ensure safety. An optimization-based planner, the BoundMPC framework [1], is utilized to execute optimal, safe, and collision-free trajectories along the reference path. The effectiveness of the approach is shown on various everyday manipulation tasks in simulation and experiment, shown in the video at www.acin.tuwien.ac.at/42d2.

**Keywords:** Vision Language Models, Manipulation Planning, Path-following MPC

## 1 Introduction

The rise of large language models (LLMs) is not just a technological advancement, but a significant step towards enabling robots to understand and reason about the world. This development has practical implications, as it allows for the simplified use of robots in everyday tasks and unseen environments. Non-expert users can now control robots using natural language, which is a major stride in human-robot interactions. This technology is particularly important in enabling robots to cooperate and help humans, such as in household tasks or robot-aided medical care. The generalization capabilities of LLMs simplify the process of conveying instructions from a user to a robot, which should act according to these instructions. However, transforming this language understanding into safe, performant robotic motions remains an open topic with many possible solution approaches.

This research tackles the crucial issue of translating human-generated instructions into safe and executable robotic actions. Previous studies have proposed various methods, including lexical analysis [2], step-by-step language instructions [3, 4], translation of language instructions into program code [5, 6, 7], and end-to-end learning for low-level robot actions [8]. While these approaches can accomplish the given tasks, they often fall short in terms of robot motion performance, leading to jerky movements [5, 8, 9] or reliance on predefined motion primitives [10, 4], which restricts robot motions to a predefined set of skills. To overcome these limitations, [11, 12] suggests learning discrete Cartesian waypoints and utilizing existing robot motion planners to traverse them. However, this necessitates training a neural network for each task, which is computationally intensive and lacks generalizability across different tasks. Moreover, the waypoint sequences require training data, which is challenging to provide. This research addresses these issues by proposing a modular motion planner that seamlessly integrates with large language models, resulting in smooth and performant robot trajectories that generalize well across tasks.

8th Conference on Robot Learning (CoRL 2024), Munich, Germany.

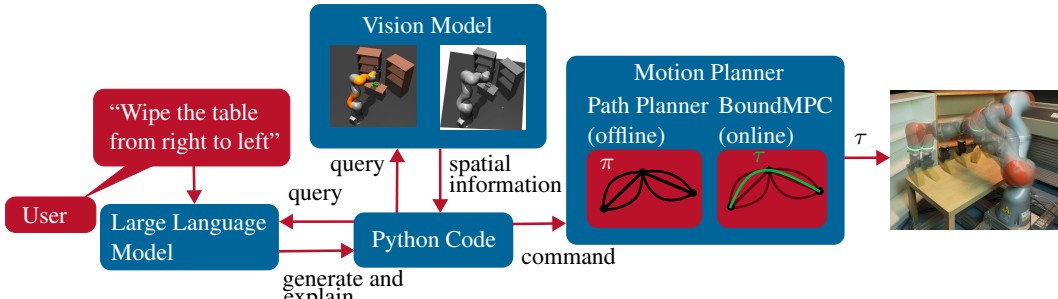

Figure 1: Overview of proposed modular motion planning framework.

The controllers of robot manipulators operate at high-frequency ($\geq 1\,\text{kHz}$) to control the input torques of the robot joints. In contrast, human-generated instructions are high-level plans provided at a significantly lower frequency.

Bridging this gap between high-level plans and low-level control requires considering various cognitive, visual, spatial, and robotic requirements. Our approach tackles this challenge with a modular architecture, see Fig. 1. This architecture includes an LLM to interpret human instructions, a visual model to localize and reason about objects in the scene, a 3D information model to exploit spatial data, a path planner to create Cartesian reference paths with collision-free bounds, and the model predictive path-following concept BoundMPC [1] to execute a safe and efficient robot trajectory along the reference path.

The generalization capabilities of the LLM can translate complex human-generated instructions into program code to communicate robot behavior to the underlying modules. However, it cannot directly deal with collision avoidance or the robot's kinematic and dynamic constraints. The LLM instructs a path planner to find feasible collision-free paths toward different goals, like objects or locations in the scene, which are specified from a zero-shot visual model [13]. The 3D information of the scene is used to construct a graph of convex collision-free sets, which is the basis for planning a Cartesian reference path. Based on these sets, the bounds around the reference paths are defined to avoid collisions with the environment. The predictive path-following controller BoundMPC [1] guides the robot towards the goal. BoundMPC computes and executes locally optimal trajectories in real-time to follow the reference path within the given bounds. Splitting path and trajectory planning has the advantage that no global trajectory optimization is needed, allowing shorter planning times. The convex-set-based path planner allows simple interaction with the generalization capabilities of the LLM, and the optimization-based trajectory planner BoundMPC creates fast and smooth trajectories.

The main contribution of this work is a novel modular motion planning pipeline. This pipeline comprises a path planner based on convex sets, which is a unique approach, as well as the interfaces of this planner with an overlaying learning-based task planner, i.e., a pre-trained LLM and a pre-trained vision model, and the underlying model predictive trajectory planner BoundMPC. This unique combination allows reasoning in Cartesian space by leveraging the strength of learning-based models deeply integrated with smooth and safe optimization-based motion planning in the joint space. The capability to create smooth, performant trajectories for unseen tasks is demonstrated in simulation and experiment by using a robot manipulator, showcasing the effectiveness of this approach. The main novelties of the proposed approach can be summarized as follows:

1. The modular motion planning framework can plan smooth, safe, and performant motions based on natural language instructions. This framework does not rely on motion primitives but generates via-points and task-specific path constraints.
2. A convex-set-based Cartesian path planner with Cartesian bounding functions describes the obstacle-free region around the path. This builds a crucial component for interfacing the learning-based task planner with the optimization-based trajectory planner BoundMPC.

## 2 Related Works

The proposed work falls in **task and motion planning (TAMP)** [14, 15]. TAMP is concerned with breaking down a complex task into, e.g., motion instructions for a robot [16, 17, 18]. Specifically, this work deals with the manipulation among movable objects planning [19]. A recent and important subfield is **language-based motion planning**, where tasks are specified as natural language instructions and are translated to executable robot trajectories. Task understanding and ordering are assumed to happen within the language model. Extracting information from language has been studied extensively for general applications [20, 21, 22] and specifically for robots [23, 3, 24, 25, 2, 26]. These language-grounding tools can be applied to many problems, such as learning reward functions in reinforcement learning [27, 28], instructing intelligent agents to obtain action policies [29], or communicating domain knowledge [30]. This work focuses on grounding task instructions for robotic manipulation tasks [5, 9]. In order to successfully plan motions for a manipulator, the language instruction has to be grounded with visual information. The authors in [24] use a textual description of the scene within a language model. More advanced works [31, 25, 12, 32] use image embeddings in combination with language embeddings as inputs for a network. Training these networks for robotic tasks is computationally expensive and requires large datasets. Another approach combines an LLM for language understanding and a vision model to correlate the language features with the objects in the scene [5, 33, 34]. Since the language and vision models are pre-trained on large general datasets, no additional training is needed, which makes the motion planner inherit the generalization capabilities and zero-shot task understanding of the pre-trained networks without any computational costs for training. Furthermore, the language model programs (LMPs) [6] demonstrate strong coding capabilities, directly translating the natural language input into Python code. However, these works lead to jerky and slow robot motions and do not consider the robot kinematics for motion planning. These undesired motions occur in end-to-end learning frameworks [8] but also for networks outputting discrete actions [5, 9] due to insufficient lower-level motion generation. In contrast, our approach uses a path planner that systematically integrates with the language model and outputs task-constrained Cartesian bounded reference paths executed with the trajectory planner BoundMPC [1] for fast, safe, and smooth motions with bounded path deviations, accounting for the robot kinematics. Safety is an important feature in motion planning and is often addressed by relying on underlying planners that guarantee safety. The work on drone swarms in [35] uses an underlying safety filter, and the manipulation planning framework in [11] uses existing joint-space motion planners with obstacle avoidance. The trajectory planner BoundMPC [1] fulfills a similar role in this work. In addition to safety, it also allows the incorporation of task-specific Cartesian constraints, e.g., keeping a cup upright or moving along a straight line.

**Manipulator trajectory planning** must consider the robot's kinematics, dynamics, and interaction with the environment. Integrating these capabilities directly into LLM-based reasoning is infeasible and does not generalize across tasks. Due to this complexity, different methods for trajectory planning have emerged, which are based on sampling [36, 37], optimization [38, 1], learning [39, 40], or a combination of multiple methods [11]. Obstacles avoidance is commonly realized by parametrizing the space occupied by obstacles and planning in the remaining space [38, 37, 5]. On the contrary, planning based on convex sets finds convex obstacles-free regions and plans a trajectory within these regions [41, 42]. The important difference between these two approaches is that the convex regions do not necessarily cover the whole obstacle-free space but still allow task completion by finding a path through the environment. The work [43] navigates through sets of convex regions using a graph, which is insufficient for motion planning. The works [42, 41, 43] directly optimize a trajectory over the path of sets. This is computationally expensive for robotic Cartesian task-space trajectories. Therefore, we propose a novel Cartesian path planner, which computes a Cartesian reference path inside connected convex regions and determines bounds of obstacle-free space for a robot manipulator to move inside. Planning a path through a graph of convex sets can be done using piecewise linear paths where each linear segment is contained in a convex set. The convexity of the set guarantees that the whole path is within the convex sets. Due to their discontinuities, existing path-following controllers, such as [44, 45, 46], struggle to follow such piecewise linear reference

paths. In this work, a novel path planner based on Cartesian convex sets is proposed and combined with the path-following controller [1]. An alternative approach is planning a global offline trajectory [36, 37, 47, 48, 49, 50, 51], which is computationally expensive. Furthermore, our path planner provides an intuitive interface with natural language instructions and enables an easy specification of task-specific constraints, while the online trajectory planner handles the robot's kinematics and motion speed. This is computationally more efficient and allows for online adaptions.

## 3   Safe Language-based Trajectory Generation

This paper solves the problem of generating a robot manipulator trajectory $\tau$ in joint space using a language instruction $\mathcal{L}$ given by a user, e.g., "Put the cup on the table". Furthermore, the robot must perform this motion safely such that the joint-space trajectory is contained within the set of safe trajectories $\tau \in \tau_{\text{safe}}$. This can be defined as

$$\underset{\tau \, \in \, \tau_{\text{safe}}}{\text{find}} \, \tau \quad \text{s.t.} \, I_{\text{success}}(\mathcal{L}) = 1 \,, \tag{1}$$

where the indicator function $I_{\text{success}}$ indicates a successful task execution. This problem formulation is general and applicable to many different tasks. The generality makes solving the problem difficult as many solutions may exist. Directly searching over $\tau_{\text{safe}}$ is infeasible as there is a significantly large number of feasible trajectories in general.

Therefore, the problem (1) is subdivided into subproblems. Thus, similar to [5], it is assumed that the instruction $\mathcal{L}$ consists of $n$ subtasks $l_i$ such that $\mathcal{L} = \{l_1, l_2, \ldots, l_n\}$, e.g., the instruction "Put the cup on the table" may be subdivided into the subtasks "Detect the cup", "Move to the cup", "Grasp the cup", "Move to the table", and "Release the cup". Each of the subtasks $l_i$ is successfully performed by a corresponding trajectory $\tau_i$. In this work, each trajectory $\tau_i$ is represented by a set of via-points $\mathcal{W}_j = \{\mathbf{w}_{\text{via},j}, \, j = 1, \ldots, m\}$, where each via-point consists of a desired end-effector pose $\mathbf{p}_{\text{via},j}$, containing position and orientation, and a gripper state $g_{\text{via},j} \in [0, 1]$, corresponding to closed ($g_{\text{via},j} = 1$) or open ($g_{\text{via},j} = 0$). Additionally, the collision-free space along the path between the via-points is specified using bounds to ensure collision-free trajectories $\tau_i \in \tau_{\text{safe}}$.

### 3.1   Motion Planning Framework

This section presents our developed motion planning framework for solving problem (1). An overview of the framework is shown in Fig. 1. It consists of four distinct components, namely:

1. An LLM for breaking down the instructions $\mathcal{L}$ into actions $\mathcal{A}$ defined in Python code,
2. a vision model to align the actions $\mathcal{A}$ of the LLM with the RGB image of the scene and retrieve spatial locations from a 3D point cloud,
3. a novel path planner based on convex sets using the IRIS algorithm [52] to plan a piecewise linear bounded and collision-free reference path $\pi$,
4. and the model predictive controller BoundMPC [1] to traverse the reference path $\pi$.

The LLM is instructed to output Python code to perform the necessary actions $\mathcal{A}$, see Fig. 1. This work uses a combination of language model programs (LMPs) [6] with Chat-GPT4 [53]. To this end, the generated Python code uses the 2D vision model and the LLM to retrieve specific objects and locations from the scene. The corresponding 3D coordinates are obtained by querying the 3D point cloud. In order to execute action $\mathcal{A}$ on the robot, the motion planner is employed to move the robot's end-effector to the planned location. In turn, the path planner first determines an obstacle-free path and bounds using convex sets of the scene. Finally, the model predictive path-following controller BoundMPC executes the robot motion. The first three steps are performed offline, and the final trajectory planning using BoundMPC is performed online at $10\,\text{Hz}$. The planned trajectories are sent to a computed-torque controller that computes the necessary joint torques at $8\,\text{kHz}$.

This idea is based on the work in [6] also used in [5] for robot motion planning. The main difference to this work is the trajectory planning. The work in [5] uses smoothed 3D cost maps to plan the end-effector trajectory, which is not guaranteed to be executable or collision-free. Furthermore,

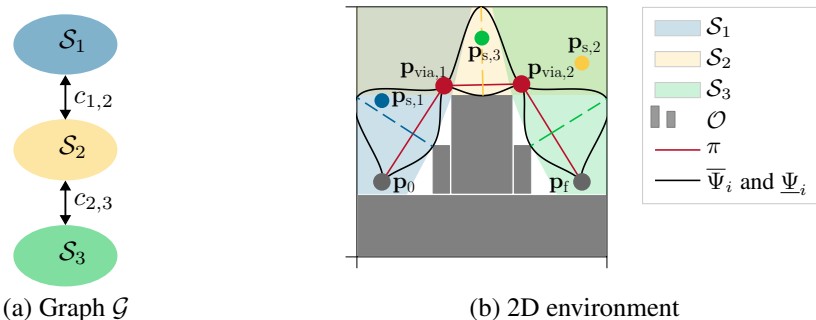

| | |
|---|---|
| (a) Graph $\mathcal{G}$ | (b) 2D environment |

Figure 2: 2D planning example using the convex-set-based path planner. (a) graph $\mathcal{G}$ of convex sets $\mathcal{S}_i$, (b) environment for planning and the resulting reference path $\pi$.

the Cartesian trajectories are converted to joint-space trajectories by sending the desired Cartesian end-effector pose of the current time step to a controller. This formulation leads to jerky and slow robot motions. The following further explains the individual components of the proposed modular framework.

### 3.1.1 Vision Model

Performing a task requires the robot to understand the surrounding scene visually. The vision model comprises the zero-shot open-vocabulary object detection network OWL-ViT [13]. The model is queried to detect objects of a specific kind, e.g., tables, shelves, and cups, and returns a list of objects with the corresponding areas in the RGB image that make up this object. After refining the 2D areas using the semantic segmentation network SegmentAnything [54], they are projected into the 3D point cloud to locate the objects in the scene spatially.

The LLM creates Python code in this work, invoking the other components to complete the task. The code queries the vision model to get scene information. For example, when the user asks the robot to "move the cup to the table", the code will query the vision model to locate the cup and the table. If several cups are detected, the Python code will call the LLM to clarify the relevant object instance for the task. The combination of the LLM with the vision model is inspired by VoxPoser [5].

### 3.1.2 Path Planner based on Convex Sets

In order to successfully perform a task, the LLM generates robot actions that move the end-effector from the current pose $\mathbf{p}_0 = \mathbf{p}_{\text{via},0}$ to a desired pose $\mathbf{p}_f = \mathbf{p}_{\text{via},L}$. A novel path planner based on convex sets is employed to find a collision-free path in Cartesian space. The core idea of the planner is to compute collision-free convex sets in the Cartesian space [52] and connect them to a graph based on their overlaps. A graph planner [43] then finds a path through the graph, which is converted into a linear reference path with Cartesian bounds, to be executed by BoundMPC [1]. The novelty of this planner is the computation of the bounded reference path $\pi$ to traverse the graph. An alternative approach would be to compute a collision-free reference path using existing path planning methods, e.g., [36, 48]. However, this needs to consider the robot's kinematics which leads to high computational complexity. Our approach defines a bounded Cartesian space along the path in the offline step and utilizes online trajectory planning to account for the robot's kinematics. This is computationally less expensive, and it is possible to incorporate task-specific constraints specified by natural language during the reference path computation, see Appendix E for more details.

In the following, the functionality of the planner is explained in detail for an example 2D environment, see Fig. 2. The 2D environment in domain $\mathcal{D} \subset \mathbb{R}^2$ contains six convex grey obstacles that constitute the obstacle space $\mathcal{O}$. The path planner must find an obstacle-free path $\pi$ from the starting point $\mathbf{p}_0$ to the end point $\mathbf{p}_f$. The IRIS algorithm [52] is employed to compute collision-free convex sets $\mathcal{S}_k \in \mathcal{D} \backslash \mathcal{O}$ around the sample points $\mathcal{P} = \{\mathbf{p}_{\text{s},k} : k = 1, \ldots, K\}$. Each set is represented as a linear inequality $\mathcal{S}_k = \{\mathbf{p} \in \mathcal{D} \backslash \mathcal{O} : \mathbf{A}_k \mathbf{p} \leq \mathbf{b}_k\}$ with the matrix $\mathbf{A}_k$ and the vector $\mathbf{b}_k$,

describing the halfspaces that define the convex set $\mathcal{S}_k$. A graph $\mathcal{G}$ of convex sets is build as detailed in Appendix A. The optimal path through the graph $\mathcal{G}$ is given as the sequence of sets $\mathcal{S}_{\mathrm{path},l}$ with $l = 1, \ldots, L$. As the sets describe safe regions for the robot to move in, constructing a reference path is desirable to systematically consider the safe space when planning toward the endpoint $\mathbf{p}_{\mathrm{f}}$. Linear reference paths with Cartesian bounds achieve this goal, and their construction based on the sets $\mathcal{S}_{\mathrm{path},l}$ is detailed in Appendix B. The resulting piecewise linear path is shown in Fig. 2b. Linear paths have the advantage of being contained within a convex set as long as their start and end points are. This is not the case for more complex formulations such as splines.

The robot must remain within the convex sets $\mathcal{S}_{\mathrm{path},l}$ to ensure collision freedom. This is ensured using the bounding functions $\overline{\Psi}_i$ and $\underline{\Psi}_i$ around the path segment $\pi_l$ within BoundMPC. In order to interact with the path-following controller in Section 3.1.3, it is desirable to obtain smooth bounding functions, detailed in Appendix C, because BoundMPC is an optimization-based framework.

The above approach is used for the position reference path of the end-effector. Finding convex sets for the orientation is more challenging due to the more complex space of orientations in three dimensions. The current planner assumes knowledge of the orientation at the starting point $\mathbf{p}_0$ and end point $\mathbf{p}_{\mathrm{f}}$, which can be obtained from grasp pose detection, e.g., [55, 56]. Grasp pose detection is an important feature, especially for complex object geometries. Adding a grasp pose detection model to our framework is straightforward and will be part of future work. At each via-point $\mathbf{p}_{\mathrm{via},l}$, the orientation is a linear interpolation between the start and the end orientation.

**Remark.** *The physical extent of the end-effector is taken into account by increasing the size of the obstacles by the size of the end-effector. This works best with a symmetric end-effector.*

### 3.1.3 Model Predictive Path-following Control

Following the path $\pi$ from Section 3.1.2 requires a path-following controller that systematically considers the Cartesian bounds $\overline{\Psi}_i$ and $\underline{\Psi}_i$. In this work, the receeding-horizon planning BoundMPC framework [1] is utilized. It allows the end-effector of a robot manipulator to follow the Cartesian path $\pi$ with bounds. As proposed in [1], a piecewise linear reference path formulation as constructed in Section 3.1.2 provides a simplified yet performant choice for the reference path $\pi$. The bounds guide the robot along the path $\pi$ while ensuring collision freedom. Furthermore, the Cartesian path $\pi$ may be hard to follow exactly since it does not consider the kinematics of the robot. The bounds enable the robot to deviate from the path and find optimal trajectories $\mathbf{q}(t)$ within the given collision-free Cartesian space, defined by the bounding functions $\overline{\Psi}_i$ and $\underline{\Psi}_i$. The advantages over state-of-the-art MPC formulations are discussed in Appendices E.1 and E.2. Currently, the bounds ensure collision-freedom for the end-effector. However, it is straightforward to add collision-avoidance terms to enforce safety for the entire kinematic chain of the robot, as shown in Appendix F.

## 4 Experimental Results

The trajectory planning framework developed in Section 3 is evaluated in simulation and experiment in this section. As a simple example, a cup is picked up and placed on the opposite side of a book, placed upright on a table. Afterward, tasks are executed to show the proposed method's versatility, generalization, and robustness. For the implementation of the path planner based on convex sets in Section 3.1.2, the Drake software framework [57] is employed. Appendix H shows the LLM prompts used in this section, which rely on the functions listed in Appendix G. More simulation results comparing the individual components of our proposed framework with state-of-the-art methods are shown in Appendix E, and collision-avoidance examples are given in Appendix F.

The trajectory to pick up the cup for the prompt "Place the cup left to the book" is visualized in Fig. 3a. The path planner from Section 3.1.2 plans a piecewise linear reference path $\pi$ from the starting point $\mathbf{p}_0$ to the end point $\mathbf{p}_{\mathrm{f}}$ with one via-point $\mathbf{p}_{\mathrm{via},1}$. The BoundMPC framework follows $\pi$ to reach $\mathbf{p}_{\mathrm{f}}$ without colliding with the book. This is compared to VoxPoser [5], where the desired end-effector trajectory is planned based on 3D cost maps. The book and the table have a negative cost in repelling the end-effector, while the cup has a positive cost in attracting the end-effector. The

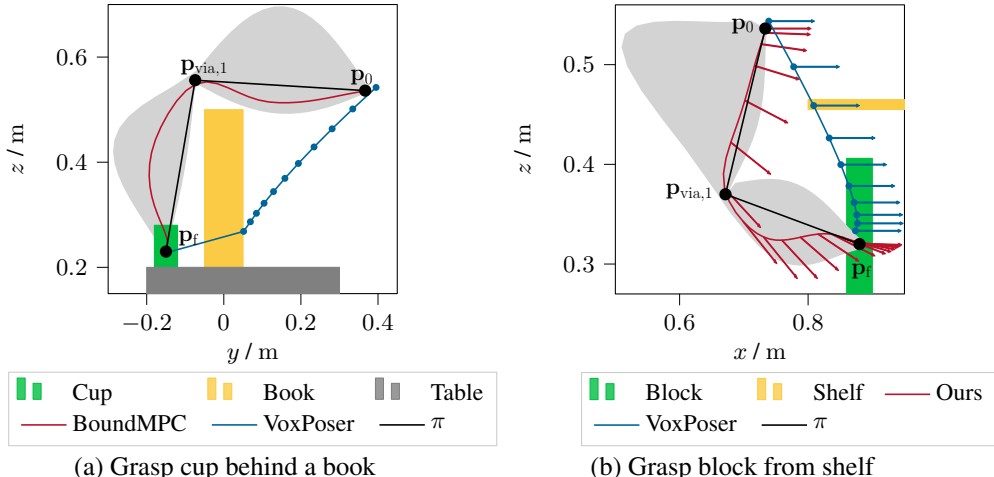

(a) Grasp cup behind a book  (b) Grasp block from shelf

Figure 3: Comparison of the end-effector trajectories for BoundMPC and the desired end-effector trajectory of VoxPoser to (a) grasp the green cup while avoiding the yellow book and the grey table and (b) grasp the green block from the yellow shelf in the environment shown in Fig. 5. The trajectories are projected into the $y$-$z$-plane and $x$-$z$-plane, respectively. The light grey shaded area is the enclosed area of the bounding functions where the robot's end-effector is allowed to move. The starting points do not coincide due to a smoothing filter applied by VoxPoser, resulting in non-smooth behavior at the beginning. The arrows in (b) indicate the end-effector's orientation around the $y$-axis, where the path bounds are chosen high to allow BoundMPC to optimize the orientation.

Table 1: Comparison of our method with VoxPoser (VP) [5] for six tasks in terms of successful task execution, collision-freedom, average path length, and intermediate stops.

| Task | Success | | Collision-free | | Avg. path length | | Stops | |
|---|---|---|---|---|---|---|---|---|
| | VoxPoser (VP) | Ours | VP | Ours | VP | Ours | VP | Ours |
| Simple Move | 33 % | 100 % | 83 % | 100 % | 1.44 m | 1.48 m | 3 | 2 |
| Swapping | 0 % | 100 % | 50 % | 100 % | - | 4.43 m | 11 | 6 |
| Reposition | 100 % | 100 % | 100 % | 100 % | 1.41 m | 0.82 m | 3 | 2 |
| Arrangement | 0 % | 100 % | 0 % | 100 % | 4.33 m | 3.72 m | 11 | 6 |
| Tea Cup | 0 % | 50 % | 100 % | 100 % | 2.11 m | 2.02 m | 3 | 2 |
| Wiping | 100 % | 100 % | 100 % | 100 % | 1.84 m | 1.22 m | 4 | 1 |

planned trajectory gets stuck on the right side of the book and then jumps to the endpoint $\mathbf{p}_f$. This results in a collision with the book. It is also not possible to tune the cost functions appropriately to achieve a collision-free trajectory using VoxPoser, as it relies on RRTConnect to create a joint trajectory from the desired Cartesian trajectory shown in Fig. 3. This procedure does not guarantee that the desired Cartesian trajectory is followed and invalidates any collision-free guarantees in the reference trajectory. BoundMPC ensures that the end-effector remains safe without any cost tuning.

The performance of our proposed methods is further evaluated in simulation in the environment shown in Fig. 1 and Fig. 5 with the following tasks:

1. **Simple Move:** "Move the [obj] [into/onto] the [left/right] shelf",
2. **Swapping:** "Swap the [obj] with [obj] by using the [table/right shelf]]",
3. **Reposition:** "Put the [obj][slowly/quickly] onto the [left/right] side of the table",
4. **Arrangement:** "Arrange the three blocks as a [line/triangle] on the table",
5. **Tea Cup:** "Place the tea cup onto the [left/right] shelf without spilling it",
6. **Wiping:** "Wipe the table from [left/right] to [right/left]",

where [obj] is chosen among [green block/blue block/yellow block]. This leads to a total of 42 distinct tasks. The simulation assumes perfect visual information such that the performance of the Python code created by the LLM is evaluated independently of the performance of the visual

model. Table 1 shows that the proposed method outperforms VoxPoser across all tasks. The **Simple Move** task is challenging for VoxPoser due to the collision avoidance where a situation similar to picking up the cup in Fig. 3 is encountered. The proposed method plans within collision-free convex sets to avoid any collisions. In the **Swapping** task, the LLM used in VoxPoser returns invalid instructions, which do not solve the task. This is caused by the independence of the subtask execution in VoxPoser, i.e., each subtask $l_i$ of the instruction $\mathcal{L}$ is assumed to be independent of all other subtasks. This assumption is invalid for the swapping scenario, as the first object must be removed, and the second object has to be moved to the previous position of the first object, which is not considered in this subtask. The **Reposition** task repositions an object on the table. Both methods perform this task successfully, but the proposed framework does this more efficiently regarding the path length. In the **Arrangement** task, the block must be grasped from different heights on the shelf and arranged on the table. VoxPoser creates a Cartesian path that the robot cannot follow due to its joint limits, which is visualized in Fig. 3b. Using our method, the bounded deviations from the path allows BoundMPC to utilize the orientation around the $y$-axis and find a suitable trajectory while avoiding the joint limits. This emphasizes the advantage of considering the robot's kinematic limits during planning and execution. The computed trajectory from VoxPoser leads to a collision with the shelf and requires a constant orientation, which the robot cannot achieve. A similar situation occurs in the **Tea Cup** task where a tea cup must be moved to a new position without spilling the tea. VoxPoser outputs a non-executable Cartesian trajectory. The LLM instructs our planner to keep the tea cup upright. BoundMPC then exploits the unconstrained rotation direction around the $z$-axis and the path deviations regarding the position to find a suitable trajectory to accomplish the task. Lastly, the **Wiping** task is solved by both frameworks. In this task, our method requires fewer stops compared to VoxPoser. The task requires picking up the sponge and moving it to one side of the table, from which the wiping motion starts. VoxPoser stops during the pick-up of the sponge, which is necessary, but also at the beginning and end of the wiping motion, which is not required. With our method, the robot moves smoothly without stopping at the via-points, leading to faster task execution. Furthermore, our method specifies that the wiping segment is constrained in position and orientation to follow the line and stay orthogonal to the table top. The LLM effectively instructs the planner to consider these constraints. Especially in the **Tea Cup** and **Wiping** tasks, the constraints specified by the LLM show a deep integration of the learning-based model and the motion planner. The LLM specifies the task-relevant constraints, and the planner finds executable, smooth, and collision-free trajectories that solve the task. The proposed framework performs well in the tasks and outperforms VoxPoser regarding task success, collision avoidance, average path length, and the number of intermediate stops.

We also demonstrate the performance of the developed framework in several experiments in the video at `www.acin.tuwien.ac.at/42d2`. The video showcases the smooth trajectories and the planner's generalizability while staying safe. More information is given in Appendix D.

## 5  Conclusion, Limitations and Future Work

This paper presents a modular language-driven zero-shot motion planning framework to create safe and smooth manipulator trajectories. A novel Cartesian path planner is developed to interface the learning-models with the trajectory generator. This allows the learning-based components to reason in Cartesian space while smooth and safe trajectories that respect the robot's kinematic limits are generated in the joint space. The strength of the proposed framework is validated in experiments and simulations. However, some limitations exists. Prompt engineering is needed to query the LLM successfully, which requires the user to invest upfront time. Furthermore, the modularity of the approach requires suitable interfaces between the modules, which can be improved for more diverse scenarios. This is especially important for the LLM and the vision module, where recent work on vision language models can provide a helpful extension. The safety of the current approach is limited by the discretization within BoundMPC [1]. In the literature, continuous-time methods guarantee safety for the whole trajectory, e.g., based on reachability [58, 59, 60]. However, in practice, we never encountered collisions due to the discretization.

**Acknowledgments**

We would like to thank Alexander Wachter for the help regarding the implementation of the comparisons and Florian Beck for the fruitful discussions.

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

## A  Graph of convex sets

The convex sets needed to build a graph $\mathcal{G}$ are computed using the IRIS algorithm [52]. The sample points $\mathcal{P} = \{\mathbf{p}_{s,k} : k = 1, \ldots, K\}$ are hyperparameters of this algorithm. In this work, the sample points $\mathcal{P}$ include the starting and end points $\mathbf{p}_0$ and $\mathbf{p}_f$, and six additional known collision-free points.

**Remark.** *Providing a small set of known obstacle-free points is often easily possible. If this is not the case, a rejection sampling approach can be adopted. More complex strategies based on the knowledge of $\mathbf{p}_0$ and $\mathbf{p}_f$ can be developed but are not part of this work. The authors of [41] propose to adapt methods developed for probabilistic-random-tree-methods to improve the sampling.*

Once collision-free convex sets are found around all sample points $\mathcal{P}$, the graph $\mathcal{G}$ is constructed in which each convex set $\mathcal{S}_k$ is represented by a node, similar to [41]. For each pair of convex sets $\mathcal{S}_a$ and $\mathcal{S}_b$ found by the IRIS algorithm, an edge is added in graph $\mathcal{G}$ between the corresponding nodes, if their intersection is not empty, i.e., $\mathcal{S}_{a,b} = \mathcal{S}_a \cap \mathcal{S}_b \neq \emptyset$. The cost associated with this edge is chosen as

$$c_{a,b} = \min_{\mathbf{p} \in \mathcal{S}_{a,b}} \|\mathbf{p} - \mathbf{p}_f\|_2^2 , \tag{2}$$

which is the projection of the end point $\mathbf{p}_f$ onto the intersection set $\mathcal{S}_{a,b}$. Note that the graph is undirected as $c_{a,b} = c_{b,a}$, and each edge allows passing in both directions. The constructed graph $\mathcal{G}$ for the 2D environment is shown in Fig. 2a. The shortest path through the convex sets is then calculated using the optimizer proposed in [43]. The optimal path through the graph $\mathcal{G}$ according to the cost (2) is given as the sequence of sets $\mathcal{S}_{\text{path},l}$ with $l = 1, \ldots, L$.

## B  Piecewise linear reference path

For constructing the piecewise linear reference path $\pi$, the sequence of sets $\mathcal{S}_{\text{path},l}$ is used. Each set intersection $\mathcal{S}_{\text{inter},l} = \mathcal{S}_{\text{path},l} \cap \mathcal{S}_{\text{path},l+1}$ with $l = 1, \ldots, L - 1$ between two consecutive sets $\mathcal{S}_{\text{path},l}$ and $\mathcal{S}_{\text{path},l+1}$ in the solution path requires one via-point $\mathbf{p}_{\text{via},l}$. The $L - 1$ intersection sets $\mathcal{S}_{\text{inter},l}$ are used to compute $L - 1$ via-points by solving the optimization problem

$$\mathbf{p}_{\text{via},l} = \underset{\mathbf{p} \in \mathcal{S}_{\text{inter},l}}{\arg\min} \|\mathbf{p} - \mathbf{p}_0\|_2^2 + \|\mathbf{p} - \mathbf{p}_f\|_2^2 + \alpha \|\max(\mathbf{A}_{\text{inter},l}\mathbf{p} - \mathbf{b}_{\text{inter},l}) + \beta\|_2^2 \tag{3}$$

with the weights $\alpha \geq 0$ and $\beta \geq 0$. Together with the starting point $\mathbf{p}_0 = \mathbf{p}_{\text{via},0}$ and the end point $\mathbf{p}_f = \mathbf{p}_{\text{via},L}$, they make up the set of via-points. In (3), the matrix $\mathbf{A}_{\text{inter},l}$ and the vector $\mathbf{b}_{\text{inter},l}$ define the intersection set $\mathcal{S}_{\text{inter},l}$, and the term $\|\max(\mathbf{A}_{\text{inter},l}\mathbf{p} - \mathbf{b}_{\text{inter},l}) + \beta\|_2^2$, with the maximum taken over the vector entries, incentivises the via-point $\mathbf{p}_{\text{via},l}$ to be $\beta$ away from any border of the intersection set $\mathcal{S}_{\text{inter},l}$. This moves $\mathbf{p}_{\text{via},l}$ from the border of the set into the set and leads to more symmetric bounds later. The piecewise linear reference path $\pi$ is the linear connection between the starting point $\mathbf{p}_0 = \mathbf{p}_{\text{via},0}$, the via-points $\mathbf{p}_{\text{via},l}$ $l = 1, \ldots, L - 1$, and the end point $\mathbf{p}_f = \mathbf{p}_{\text{via},L}$, as shown in Fig. 2. This formulation ensures that the linear segment between $\mathbf{p}_{\text{via},l-1}$ and $\mathbf{p}_{\text{via},l}$ is completely contained in the convex set $\mathcal{S}_{\text{path},l}$, from which the bounding functions are computed using geometric considerations.

## C  Path bounding functions

In the following, the computation of the bounding functions in the general case of three dimensions is described for a given linear path segment $\pi_l$, which connects the via-points $\mathbf{p}_{\text{via},l}$ and $\mathbf{p}_{\text{via},l+1}$ and is contained in the corresponding convex set

$$\mathcal{S}_{\text{path},l} = \{\mathbf{p} : \mathbf{A}_{\text{path},l}\mathbf{p} \leq \mathbf{b}_{\text{path},l}\} . \tag{4}$$

A visual explanation of the procedure is shown in Fig. 4. This procedure also applies analogously to the special case of two dimensions and the example in Fig. 2. In 3D, each linear path segment has a constant unit-norm direction vector $\mathbf{d}$ and two unit-norm basis vectors $\mathbf{b}_1$ and $\mathbf{b}_2$. These basis

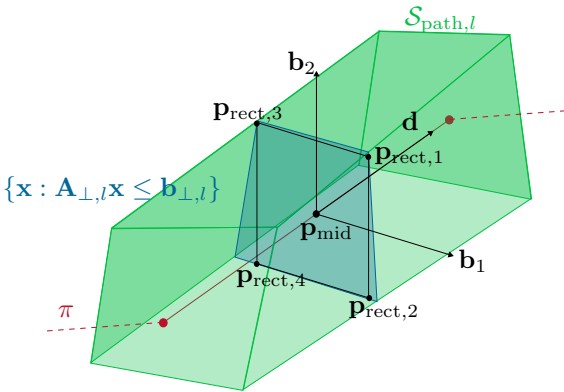

Figure 4: Computation of reference path bounding functions $\overline{\Psi}_i$ and $\underline{\Psi}_i$ based on the convex set $\mathcal{S}_{\mathrm{path},l}$ and the path $\pi$.

vectors are obtained using the Gram-Schmidt procedure explained in [1], such that an orthonormal frame is created, i.e., $\mathbf{b}_1 \perp \mathbf{d}$, $\mathbf{b}_2 \perp \mathbf{d}$, and $\mathbf{b}_1 \perp \mathbf{b}_2$. Furthermore, the mid point of the considered linear segment $\pi_l$ is $\mathbf{p}_{\mathrm{mid}}$. Next, $\mathcal{S}_{\mathrm{path},l}$ in (4) is transformed into the basis vector coordinates at the point $\mathbf{p}_{\mathrm{mid}}$ by using the transformation $\mathbf{z} = \mathbf{P}(\mathbf{p} - \mathbf{p}_{\mathrm{mid}}) = [\mathbf{b}_1 \quad \mathbf{b}_2]^{\mathrm{T}} (\mathbf{p} - \mathbf{p}_{\mathrm{mid}})$ as

$$\mathcal{S}_{\mathrm{trans},l} = \{\mathbf{z} : \mathbf{A}_{\mathrm{path},l}\mathbf{P}^{\mathrm{T}}\mathbf{z} \leq \mathbf{b}_{\mathrm{path},l} - \mathbf{A}_{\mathrm{path},l}\mathbf{p}_{\mathrm{mid}}\} \, . \tag{5}$$

A rectangle that is aligned with the basis vectors $\mathbf{b}_1$ and $\mathbf{b}_2$ in the transformed set (5) is defined by the four corner points $\mathbf{z}_{\mathrm{rect},1}^{\mathrm{T}} = [x_{1,1}, x_{2,1}]$, $\mathbf{z}_{\mathrm{rect},2}^{\mathrm{T}} = [-x_{1,2}, x_{2,1}]$, $\mathbf{z}_{\mathrm{rect},3}^{\mathrm{T}} = [x_{1,1}, -x_{2,2}]$, and $\mathbf{z}_{\mathrm{rect},4}^{\mathrm{T}} = [-x_{1,2}, -x_{2,2}]$ with the parameter vector $\mathbf{x}^{\mathrm{T}} = [x_{1,1}, x_{1,2}, x_{2,1}, x_{2,2}]$. The bounds are computed by finding the largest-volume rectangle that is contained in the set $\mathcal{S}_{\mathrm{trans},l}$ using the optimization

$$\mathbf{x}_{\mathrm{opt}} = \arg\max_{\mathbf{x}}(x_{1,1} + x_{1,2})(x_{2,1} + x_{2,2})$$
$$\text{s.t. } \mathbf{x} \geq \mathbf{0} \, , \tag{6}$$
$$\mathbf{z}_{\mathrm{rect},i} \in \mathcal{S}_{\mathrm{trans},l} \, , i = 1, \ldots, 4 \, ,$$

where the constraints ensure that the rectangle is contained in the transformed set (5) and that the path $\pi$ is contained in the bounds at $\mathbf{p}_{\mathrm{mid}}$ with $\mathbf{x} \geq \mathbf{0}$. This rectangle is generally a conservative approximation of the transformed set (5). The rectangle parameters are used as the maximum bounding function values in the linear segment $\pi_l$. For each line segment, four bounding functions exist, i.e., the upper bounding functions $\overline{\Psi}_i$ and the lower bounding functions $\underline{\Psi}_i$ for both basis directions $\mathbf{b}_1$ and $\mathbf{b}_2$. The respective maximum and minimum values are chosen as $\max(\overline{\Psi}_i) = x_{i,1}$ and $\min(\underline{\Psi}_i) = -x_{i,2}$, shown as dotted lines in Fig. 2b. The value of the bounding functions at the via-points $\mathbf{p}_{\mathrm{via},l}$ is zero, which the path-following controller requires. More information on the bounding function formulation is given in [1]. When choosing the bounding function as piecewise linear functions between zero at the via-points and $\max(\overline{\Psi}_i)$ and $\min(\underline{\Psi}_i)$ at $\mathbf{p}_{\mathrm{mid}}$, it is ensured that they remain within the set $\mathcal{S}_{\mathrm{path},l}$ because it is convex. However, the path-following controller benefits from smooth bounds, so fourth-order polynomials are used in this work instead. This can result in some parts of the bounds lying outside the set $\mathcal{S}_{\mathrm{path},l}$. However, in practice, this does not cause any issues due to the conservative approximation of the collision-free space of the IRIS algorithm and the conservative approximation in (6). The bounding functions $\overline{\Psi}_i$ and $\underline{\Psi}_i$, in combination with the reference path $\pi$, are used for the trajectory generation in Section 3.1.3.

## D   Experimental Setup

The scene of the experiments is shown in Fig. 5. It consists of two shelves and a table in front of the robot. An Azure Kinect RGB-D camera is mounted behind the robot to provide the required data for the vision model, namely the RGB picture and the depth information used to obtain the point cloud. The user provides a prompt at the beginning of an episode, and the LLM creates

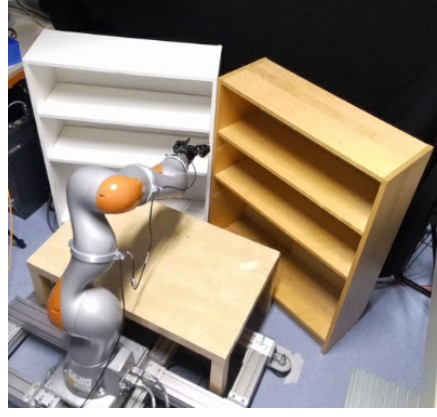
(a) Experimental scene

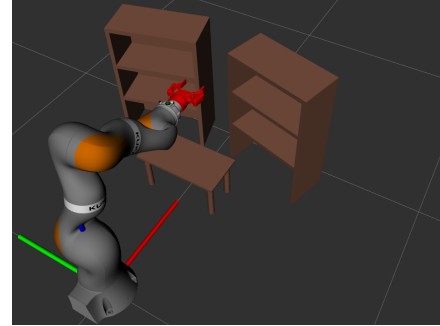
(b) Simulation scene

Figure 5: A view of the experimental and simulation scenes: The experimental scene picture is taken by the RGB-D Azure Kinect camera, also used by the vision module. The colored axes in the simulation scene at the robot's base indicate the world frame.

the necessary Python code to execute the task. BoundMPC runs at $10\,\mathrm{Hz}$, and a computed torque controller executes the created joint trajectories at $8\,\mathrm{kHz}$.

The convex set planner from Section 3.1.2 requires a convex obstacles description of the scene. For arbitrary scenes, a voxel grid based on the point cloud can be used in the planner. This may lead to many obstacles and increases the computation time of the planner, but it allows the robot to act safely in unknown environments. To speed up the computation, a known convex decomposition of the scene is employed in this work. This results in an environment with 13 obstacles, leading to computation times of about $0.15\,\mathrm{s}$ for the path planner.

Common failures of the vision model are falsely detecting an object that is not present or not detecting an object that is present. Further failure cases of the LLM and the vision model are discussed in [5].

# E   Advantages of convex-set-based path planning with trajectory optimization

This section compares the proposed motion planner, consisting of the convex-set-based path planner from Section 3.1.2 and the trajectory planner BoundMPC [1] to existing approaches. The advantages are summarized in Appendix E.4. Additionally, the simulation comparisons are shown in the video at www.acin.tuwien.ac.at/42d3.

## E.1   BoundMPC compared to Cartesian tracking MPCs (CT-MPC)

The current work uses an offline path planner in combination with the BoundMPC framework to compute suitable joint space trajectories. Other formulations exist to plan such trajectories. This section compares our proposed framework to an MPC trajectory tracking formulation [61, 62]. A reference trajectory is created for the robot, and the goal is to follow this trajectory using a receding horizon MPC while avoiding obstacles. The obstacles are commonly added as potential functions to the MPC objective [63]. A reference trajectory consists of a reference path with a time parametrization. A Cartesian reference trajectory determines the end-effector's desired position and orientation reference trajectory. This formulation is known as CT-MPC. The reference trajectory is generated based on a sequence of via-points according to the procedure described in [64] and is not guaranteed to be collision-free.

For a detailed simulation comparison, a green block is positioned in the lower shelf part inside the left shelf of the environment shown in Fig. 5. The goal is given by the prompt "Move the green

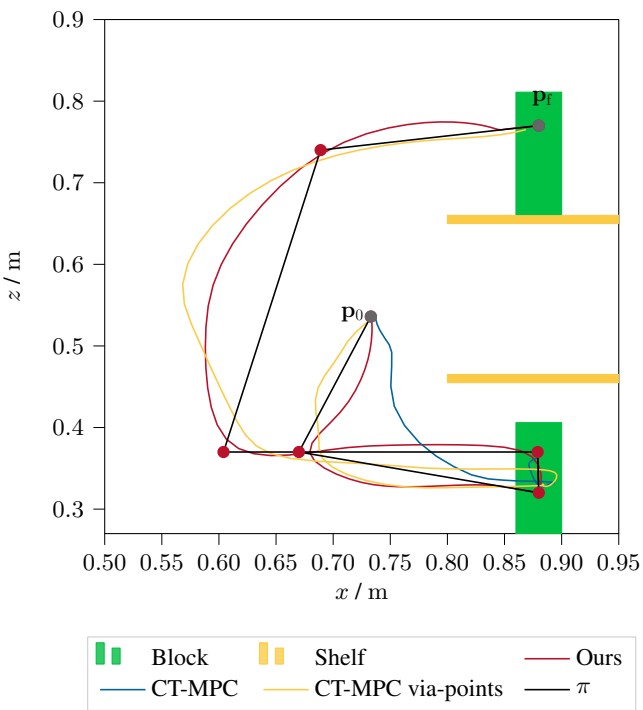

Figure 6: Comparison of trajectories of different MPC formulations for the prompt "Move the green block onto the left shelf." in the $x$-$z$-plane. This example showcases the need for a suitable reference path/trajectory to execute the task successfully.

block onto the left shelf.". The comparison of the created trajectories is shown in Fig. 6. For the simple CT-MPC formulation, only the desired grasping pose and drop off pose are provided as via-points. This does not lead to a successful task execution as the robot gets stuck on the lower shelf since it tries to move up straight but is blocked by obstacle avoidance. Thus, a more advanced reference trajectory is needed. For the CT-MPC formulation with more via-points, the via-points of the convex-set-based path planner from Section 3.1.2 are utilized to create a reference trajectory. This leads to a successful task execution with a trajectory similar to our proposed method, showing that a global reference path/trajectory is necessary for the task's success.

The CT-MPC formulation needs a reference trajectory but does not require bounds around this trajectory as obstacle avoidance is performed using potential functions in the MPC objective function. The objective of the CT-MPC is to minimize the tracking error between the end-effector pose and the reference pose. This leads to problems when the reference trajectory is not followable due to the robot's kinematics. An example is shown in Fig. 7. Here, the robot picks up a green block and has to put it on the shelf behind itself. Based on the via-points, the created reference trajectory is hard to follow because it passes very close to the robot's base's vertical axis. Hence, a large tracking error is needed to perform the task, as shown in Fig. 8b. However, the objective of CT-MPC is to minimize this error, and the robot's kinematics hinder a minimization, leading to non-smooth motions. The norm of the end-effector acceleration $\mathbf{a}_c$ reflects this behavior. Furthermore, CT-MPC does not perform the task successfully because the large reference tracking error $\mathbf{e}_{ref}$ leads to the robot getting stuck in a local minimum above the desired shelf space, which leads to a non-zero tracking error $\mathbf{e}_{ref}$ at the end of the trajectory. This is avoided by using a bounded reference path $\pi$ as this method determines which path deviations are suitable at which point in space. Hence, the proposed approach finds a smoother and successful end-effector trajectory for this task.

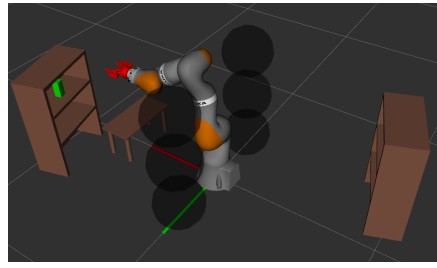
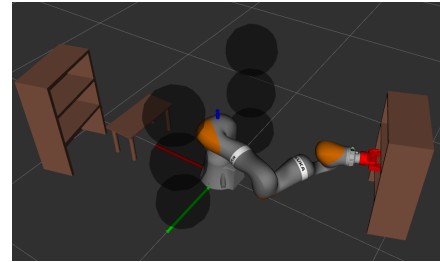

|  (a) Start position | (b) Possible end position |

Figure 7: Simulation setup for a turn-around task. The black spheres around the robot base are only used in the obstacle avoidance example in Appendix F. The starting position is the same for all formulations. Our framework can realize the end position shown. The colored axes at the robot's base indicate the world frame.

Table 2: Comparison of the offline planning time of our method with the global planners VP-STO [47], RRT* [65] and STOMP [51] for the turn-around task in Fig. 8. The planning time is the combined time for the first and second trajectory planning.

|  | Ours | VPSTO | RRT* | STOMP |
|---|---|---|---|---|
| Planning time / s | 0.3 | 53 | 30 | 411 |

### E.2 BoundMPC compared to joint-space tracking MPCs (JT-MPC)

The two comparisons above showcase pick-and-place scenarios. These can also be solved using a joint-space reference trajectory instead of a Cartesian one. We call this formulation JT-MPC. It works well for pick-and-place tasks but struggles with tasks where Cartesian task-specific constraints are given. The reference trajectory uses the inverse kinematics solutions of the via-points that result from the convex-set-based planner outputs is using. The example of the wiping task from Section 4 is considered in Fig. 9, where the wiping motion along the table surface has to be a straight line. The JT-MPC framework cannot take the Cartesian reference path $\pi$ into account, leading to an undesired arc-shaped trajectory. The proposed framework considers the task-specific constraints formulated by the LLM and successfully finishes the task. This shows the advantages of a Cartesian reference path to incorporate task-specific constraints.

### E.3 Comparison to global trajectory planning

Our approach splits the motion planning problem into offline Cartesian reference path planning and online trajectory planning. However, it is also possible to directly optimize a joint-space trajectory from the starting pose $\mathbf{p}_0$ to the final pose $\mathbf{p}_f$. In this section, the turn-around example from Appendix E.1 is considered with different global motion planners. It consists of two planned trajectories: The first trajectories grabs the green block and the second trajectory transfers it to the other shelf. Our approach is compared to VPSTO [47], RRT* [65], and STOMP [51]. A comparison of the combined planning times for both trajectories is shown in Table 2. Our approach computes a reference path $\pi$ with collision-free bounding functions considerably faster than computing a collision-free joint-space trajectory. Note that all methods manage to solve the task successfully. Furthermore, a change in the environment will require a replanning, which is costly for the global planners but not for our proposed method. This shows the advantages of using our proposed methods instead of global trajectory planners.

### E.4 Summary of advantages

The proposed approach of motion planning as a combination of convex-set-based Cartesian path planning with trajectory planning using BoundMPC has the following advantages:

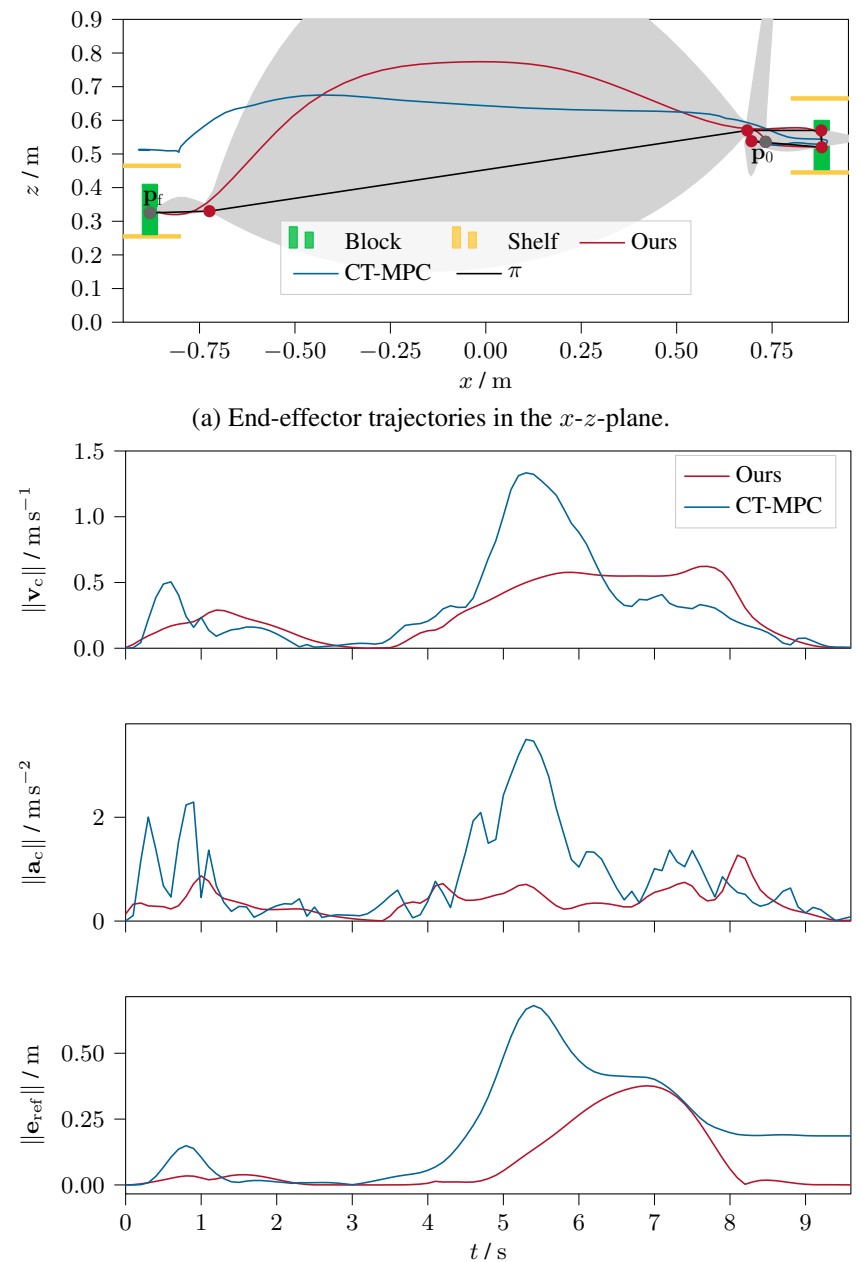

(a) End-effector trajectories in the $x$-$z$-plane.

(b) Norms of end-effector velocity $\mathbf{v}_c$, acceleration $\mathbf{a}_c$, and reference tracking error $\mathbf{e}_{ref}$.

Figure 8: Comparison of our proposed framework with CT-MPC for a turn-around task.

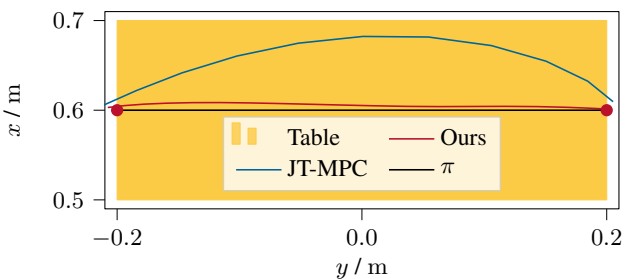

Figure 9: Comparison of trajectories of different MPC formulations for the prompt "Wipe the table from left to right." in the $y$-$x$-plane. Only the part of the trajectories relevant to the wiping is shown. This example showcases the need for a Cartesian reference path/trajectory to execute the task successfully.

- **Reference path:** A reference path/trajectory is needed to perform the task successfully and provide a global guide for the robot's motion. The example in Fig. 6 elaborates on this point.

- **Cartesian reference path:** A Cartesian reference path/trajectory has the advantage that task-specific constraints along the path are easy to incorporate compared to joint-space reference paths/trajectories, see Fig. 9.

- **Bounds around the reference path:** Existing sampling-based methods are a powerful tool to plan reference paths, e.g., RRT [66, 36] or PRM [67, 36]. However, they currently have no notion of path deviation bounding functions. The examples in Fig. 6 and 8 show the advantages of using such bounding functions. In these examples, deviating from the reference path is necessary due to the robot's kinematics, which makes minimizing the tracking error unsuitable, leading to non-smooth trajectories. The bounding functions guide the robot towards the goal while keeping it in the collision-free part of the task space. Furthermore, a Cartesian reference path is required to handle task-specific constraints, such as following a straight line exactly or keeping a cup upright. Thus, bounding functions around the reference path are a powerful tool for considering the robot's kinematics and encoding task-specific constraints.

- **Splitting up path and trajectory planning:** Our approach first computes a bounded Cartesian reference path offline and then plans a joint-space trajectory online. Another approach is to directly compute a joint-space trajectory offline using global planners, as shown in Appendix E.3. The advantages of our approach are the faster offline planning time and faster adaption in case of environmental changes.

## F  Collision avoidance for the full kinematic chain of the robot

The bounded reference path computed in Section 3.1.2 considers collisions only for the robot's end-effector. In this section, this collision avoidance is extended to the entire kinematic chain of the robot using potential functions in BoundMPC's objective function. This approach is based on the formulation in [63]. Additionally, it is compared to the Cartesian tracking (CT-MPC) and joint-space tracking (JT-MPC) MPC formulation of Appendix E.

The example in Fig. 7, where the robot has to turn around and place an object, is considered again. Six additional obstacles are positioned close to the robot base as shown by the black spheres in Fig. 7. This is a very demanding task as the space where the robot can turn around is highly constrained due to the obstacles. The resulting end-effector trajectories of our framework, CT-MPC, and JT-MPC are depicted in the $x$-$z$ and $x$-$y$-planes in Fig. 10a and b, respectively. Our framework and JT-MPC successfully perform the task, but the CT-MPC gets stuck at about $x = 0.05$ while turning around due to the obstacles. This is because the robot must deviate from the Cartesian reference trajectory due to the kinematic constraints but has no guidance on which deviation directions are suitable. Our

Table 3: Comparison of our method with CT-MPC and JT-MPC for a turn-around task with collision avoidance of the robot's full kinematic chain. Nine experiments are compared, in which the six obstacles in Fig. 7 are shifted along the $x$-axis in their $x$-coordinate between $-0.4$ and $0.4$.

|  | JT-MPC | CT-MPC | Ours |
|---|---|---|---|
| Successes | 4/9 | 3/9 | 6/9 |
| Collisions | 5/9 | 0/9 | 0/9 |
| Stuck at joint limits | 0/9 | 3/9 | 3/9 |
| Stuck at obstacles | 0/9 | 3/9 | 0/9 |

framework uses the BoundMPC formulation, which defines the Cartesian bounding functions such that the end-effector stays collision-free. It is the only formulation that can completely perform the task successfully. Even though JT-MPC reaches the goal pose $\mathbf{p}_f$, a collision of a robot link with one of the collision spheres is detected, as shown in Fig. 10c. In this example, a trade-off between minimizing the trajectory error and the collision avoidance leads to an unsafe trajectory of the robot. Our framework avoids such situations since the reference paths $\pi$ are collision-free by design. A possible improvement to JT-MPC would be providing a collision-free joint-space trajectory for the JT-MPC. However, this is not trivial and would require a global trajectory planner, which is computationally expensive, as discussed in Appendix E.3.

In order to evaluate the robustness of our proposed method, the obstacles were shifted along their $x$-coordinate in the range $-0.4$ and $0.4$. The evaluation of the resulting nine experiments is shown in Table 3. JT-MPC always reaches the desired final pose $\mathbf{p}_f$ but often collides with the environment due to the collisions in the reference trajectory. The CT-MPC formulation never collides with the environment but may get stuck at obstacles or joint limits. Our approach is the most successful framework, which only sometimes gets stuck at joint limits when the reference path bounds do not allow enough freedom to optimize a suitable joint trajectory. This comparison shows the robustness of our approach compared to the state-of-the-art.

## G   Functions used in the generated Python Code

The LLM generates Python code that can use the following functions:

- `move_to(pose: array, constraint: string, stop: bool)`
  Instructs the motion planner to move to `pose`. The argument `constraint` can be used to specify constraints along the path, which are evaluated by an LLM query, and the argument `stop` specifies whether to stop at `pose` or continue fluently towards the next motion goal.

- `detect(object_name: string) -> object`
  Invokes the vision module to detect objects with the name `object_name`.

- `close_gripper()`
  Closes the gripper.

- `open_gripper()`
  Opens the gripper.

- `move_home()`
  Move the robot's end-effector to the home position.

- `select_object(object_list: list, object_name: string, instance: string) -> object`
  Invoked in case multiple objects of the same type are detected with the `detect()` function. Queries the LLM to clarify which object is relevant to the task based on the `instance` string, e.g., finding the left shelf when `instance` is 'left' and multiple shelves are detected.

- `set_speed(max_speed: float)`
  Sets the maximum path speed for BoundMPC to be `max_speed`.

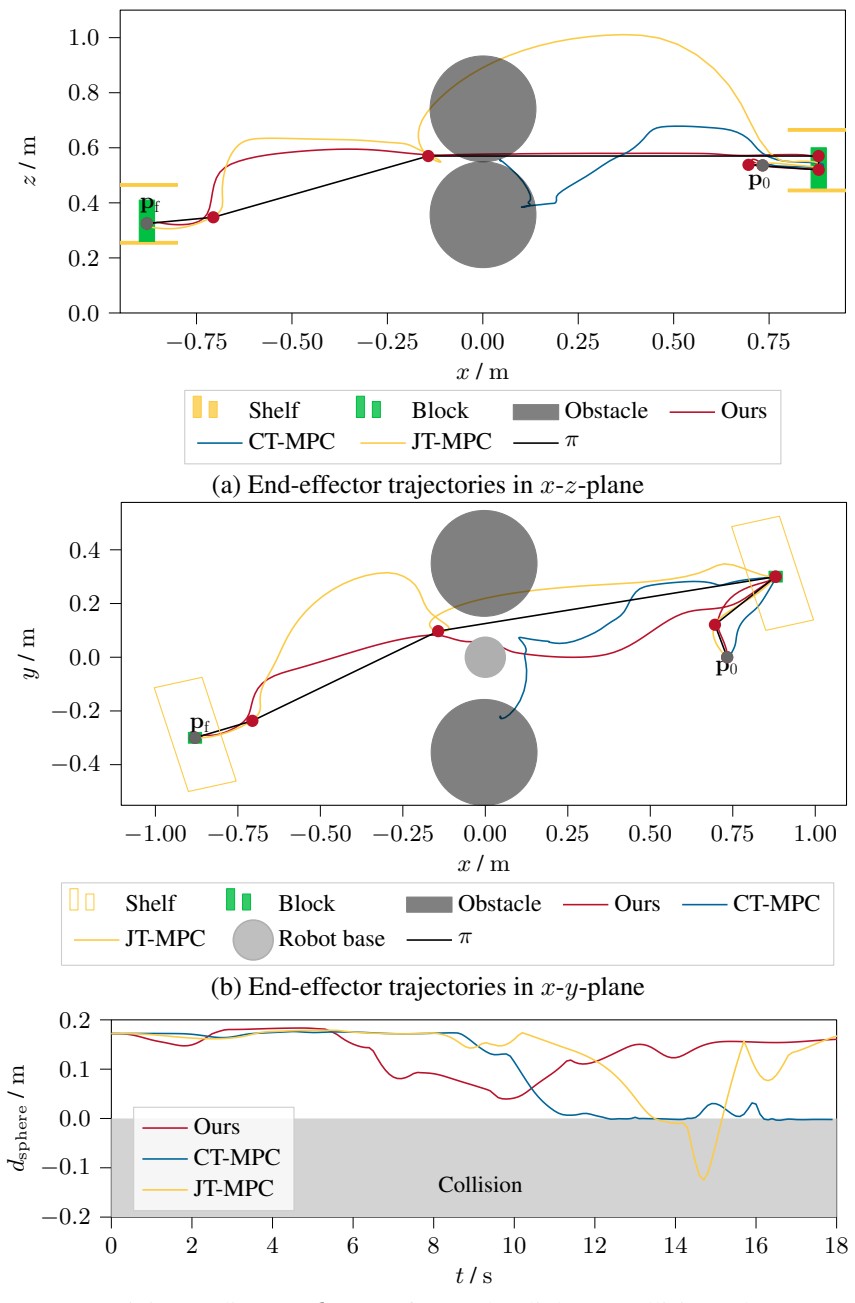

(a) End-effector trajectories in $x$-$z$-plane

(b) End-effector trajectories in $x$-$y$-plane

(c) Minimum distance $d_{\mathrm{sphere}}$ of any robot link to a collision sphere

Figure 10: Comparison of our proposed framework with CT-MPC and JT-MPC for a turn-around task with additional obstacles. The distance $d_{\mathrm{sphere}}$ is computed as the minimum distance of any robot link to one of the collision spheres and has to fulfill $d_{\mathrm{sphere}} > 0\,\mathrm{m}$ to ensure collision freedom.

- `find_position(object_move: object, object_relative: object, hint: string) -> object`
  Invokes the LLM with an object placing query. Finds a position for `object_move` relative to `object_relative` specified by the `hint`. For example, placing a cup relative to a table with the `hint` 'onto right' finds a position on top of the table to the right where the cup can be placed. The found position is saved in the return object

- `set_position(position: array)`
  Sets the position based on the adaptions from the `find_position()` function.

- `get_end_effector() -> object`
  Returns the current end-effector state of the robot.

- `empty_constraint() -> constraint`
  Returns a constraints object where all constraints are unset.

- `full_constraint() -> constraint`
  Returns a constraints object where all constraints are set.

- `set_constraint(constraint)`
  Sets the constraint for the current segment.

## H  Prompts

The LLM prompts used in this work are given in this section. The planner prompt is the highest-level prompt invoked with the natural language user input. The position prompt is used by the `find_position()` function. The constraint prompt is invoked within the `move_to()` function, and the selection prompt is used by the `select_object()` function. The selection prompt lists the objects and their center points and requires the LLM to provide an integer number of the relevant object. All other queries provide examples for the LLM, which are listed below:

Planner prompt:

```
import numpy as np
import copy
from plan_utils import (move_home, move_to, get_end_effector,
detect, open_gripper, close_gripper, select_object, set_speed,
find_position)

# Query: go back to default.
move_home()
# done

# Query: save the current pose of the plant
detected_objects = detect(["plant", "tree", "chair"])
plant = select_object(detected_objects, "plant")
saved_plant = copy.deepcopy(plant)
# done

# Query: move the three green blocks into the left shelf
detected_objects = detect(["red block", "usb stick",
    "green block", "shelf"])
shelf = select_object(detected_objects, "shelf", "left")
for i, object in enumerate(detected_objects["green block"]):
    move_to(object)
    close_gripper()
    adapted_shelf = find_position(object, relative_to=shelf,
        hint="into")
    move_to(adapted_shelf)
    open_gripper()
move_home()
```

```
# done

# Query: move the cup into the right shelf to the left
detected_objects = detect(["cup", "shelf", "plate", "glass"])
cup = select_object(detected_objects, "cup")
shelf = select_object(detected_objects, "shelf", "right")
adapted_shelf = find_position(cup, relative_to=shelf,
    hint="into left")
move_to(cup)
close_gripper()
# A cup has to be upright to not spill anything
move_to(adapted_shelf, constraint="upright")
open_gripper()
move_home()
# done

# Query: move the ball from the table into the right shelf and back
detected_objects = detect(["ball", "shelf", "block", "curtain"])
ball = select_object(detected_objects, "ball")
# Copy the start position of the ball to be able to return to it
ball_start = copy.deepcopy(ball)
shelf = select_object(detected_objects, "shelf", "right")
adapted_shelf = find_position(ball, relative_to=shelf, hint="into")
move_to(ball)
close_gripper()
move_to(adapted_shelf)
move_to(ball_start)
open_gripper()
move_home()
# done

# Query: place the block in the right shelf onto the table
detected_objects = detect(["block", "phone", "ball", "shelf",
    "table"])
block = select_object(detected_objects, "block", "right shelf")
table = select_object(detected_objects, "table")
adapted_table = find_position(block, relative_to=table,
    hint="onto")
move_to(block)
close_gripper()
move_to(adapted_table)
open_gripper()
move_home()
# done

# Query: put the trash onto the floor
detected_objects = detect(["trash", "floor", "bin"])
for object in detected_objects["trash"]:
    trash = select_object(detected_objects, "trash")
    move_to(trash)
    close_gripper()
    move_to(floor)
    open_gripper()
move_home()
# done

# Query: move the highest glass slowly to the bottom of the left
# shelf
detected_objects = detect(["glass", "shelf", "bowl", "umbrella"])
```

```
glass = select_object(detected_objects, "glass", "highest")
shelf = select_object(detected_objects, "shelf", "left")
adapted_shelf = find_position(glass, relative_to=shelf,
    hint="bottom")
move_to(glass)
close_gripper()
set_speed(0.5)
move_to(adapted_shelf)
open_gripper()
move_home()
# done

# Query: create a tower of all blue blocks on the table at the
# right side
detected_objects = detect(["blue block", "green block",
    "red block", "table"])
table = select_object(detected_objects, "table")
for i, block in enumerate(detected_objects["blue block"]):
    move_to(block)
    close_gripper()
    adapted_table = find_position(block, relative_to=table,
        hint=f"stack height {i} right")
    move_to(adapted_table)
    open_gripper()
move_home()
# done

# Query: move all books to the middle of the left shelf
detected_objects = detect(["flowers", "plate", "book", "shelf"])
shelf = select_object(detected_objects, "shelf", "left")
for book in detected_objects["book"]:
    move_to(book)
    close_gripper()
    adapted_shelf = find_position(book, relative_to=shelf,
        hint="middle")
    move_to(adapted_shelf)
    open_gripper()
move_home()
# done

# Query: move the green mug to the table and the white mug to the
# previous position of the green mug
detected_objects = detect(["mug", "lamp", "table"])
table = select_object(detected_objects, "table")
white_mug = select_object(detected_objects, "mug", "white")
green_mug = select_object(detected_objects, "mug", "green")
prev_green_mug = copy.deepcopy(green_mug)
adapted_table = find_position(white_mug, relative_to=table,
    hint="onto")
# Grasp the green mug
move_to(green_mug)
close_gripper()
# Move the green mug onto the table
move_to(adapted_table)
open_gripper()
# Move the white mug to the previous position of the green mug
move_to(white_mug)
close_gripper()
move_to(prev_green_mug)
```

```
open_gripper()
move_home()
# done

# Query: put the usb stick close to the flowers
detected_objects = detect(["usb stick", "cupboard", "flowers"])
usb = select_object(detected_objects, "usb stick")
flowers = select_object(detected_objects, "flowers")
adapted_flowers = find_position(usb, relative_to=flowers,
    hint="5cm next to")
move_to(usb)
close_gripper()
move_to(adapted_flowers)
open_gripper()
move_home()
# done

# Query: place the cup left to the coffee machine and return to
# the initial position of the cup
detected_objects = detect(["cup", "coffee machine", "table"])
cup = select_object(detected_objects, "cup")
coffee_machine = select_object(detected_objects, "coffee machine")
prev_cup = copy.deepcopy(cup) # Save the position of the cup
adapted_coffee_machine = find_position(cup,
    relative_to=coffee_machine,
    hint="10cm left")
# Grasp the cup
move_to(cup)
close_gripper()
# Move 10cm next to the coffee machine and place the cup while
# keeping it upright
move_to(adapted_coffee_machine, constraint="upright")
open_gripper()
# Move back to the initial cup position
move_to(prev_cup)
# done

# Query: set up the spoon for my soup
detected_objects = detect(["bowl", "spoon", "knife", "fork"])
bowl = select_object(detected_objects, "bowl")
# Fork and knife are not needed to each a soup
spoon = select_object(detected_objects, "spoon")
spoon_place = find_position(spoon, relative_to=bowl,
    hint="10cm right")
move_to(spoon)
close_gripper()
move_to(spoon_place)
open_gripper()
move_home()
# done

# Query: sweep all particles off the table at the left side.
detected_objects = detect(["particle", "table", "shelf"])
table = select_object(detected_objects, "table")
end_effector = get_end_effector()
for particle in detected_objects["particle"]:
    # The pre sweep position is to the right of the particle to be
    # able to push it to the left
    pre_sweep = find_position(end_effector, relative_to=particle,
```

```
            hint="3cm right")
      post_sweep = find_position(end_effector, relative_to=table,
          hint="left side")
      # Move to the pre sweep position, stopping there is not
      # necessary
      move_to(pre_sweep, stop=False)
      # During the sweep the constraint the movement to be exact to
      # follow a straight line to the post sweep position
      move_to(post_sweep, constraint="exact")
move_home()
# done

# Query: push close the topmost drawer.
detected_objects = detect(["drawer", "table", "shelf"])
topmost_drawer = select_object(detected_objects, "drawer",
    "topmost")
end_effector = get_end_effector()
close_drawer = find_position(end_effector,
    relative_to=topmost_drawer,
    hint="30cm push")
move_to(topmost_drawer, stop=False)
move_to(close_drawer, constraint="exact")
move_home()
# done

# Query: wipe the blackboard from left to right
detected_objects = detect(["sponge", "blackboard", "shelf"])
sponge = select_object(detected_objects, "sponge")
blackboard = select_object(detected_objects, "blackboard")
pre_wipe = find_position(sponge, relative_to=blackboard,
    hint="left side")
post_wipe = find_position(sponge, relative_to=blackboard,
    hint="right side")
# Pick up the sponge
move_to(sponge)
close_gripper()
# Wipe the blackboard
move_to(pre_wipe, stop=False)
move_to(post_wipe, constraint="exact")
move_home()
# done
```

Position prompt:

```
import numpy as np
from adapt_utils import set_position

# Given data:
# - hint: hint for the desired position on goal object / string

# Query: "cup" to "right shelf", hint: "onto"
cup = objects.movable
shelf = objects.goal
(max_x, max_y, max_z) = shelf.max_bounds
# Onto means the maximum z plus half the size of the object
x = shelf.center[0]
y = shelf.center[1]
z = max_z + cup.size[2]/2
set_position([x, y, z])
```

```
# done

# Query: "cube" to "shelf", hint: "into"
cube = objects.movable
shelf = objects.goal
# Into means we use the center position
set_position(shelf.center)
# done

# Query: "book" to "table", hint: "top"
book = objects.movable
table = objects.goal
(max_x, max_y, max_z) = table.max_bounds
x = table.center[0]
y = table.center[1]
z = max_z + book.size[2]/2
set_position([x, y, z])
# done

# Query: "mug" to "flowers", hint: "10cm left"
flowers = objects.goal
(max_x, max_y, max_z) = flower.max_bounds
# Add 10cm to the left border of the flower due to the "10cm left"
# hint
x = flowers.center[0]
y = max_y + 0.1
z = flowers.center[2]
set_position([x, y, z])
# done

# Query: "block" to "floor", hint: "stack height 1"
# First stack height means we add only the size of the block in
# z direction
block = objects.movable
floor = objects.goal
x = floor.center[0]
y = floor.center[1]
z = floor.max_bounds[2] + block.size[2]/2
set_position([x, y, z])

# Query: "block" to "floor", hint: "stack height 2"
# Second stack height means we add twice in z direction
block = objects.movable
floor = objects.goal
x = floor.center[0]
y = floor.center[1]
z = floor.max_bounds[2] + 1.5 * block.size[2]
set_position([x, y, z])
# done

# Query: "picture" to "mirror", hint: "in front"
mirror = objects.goal
x = mirror.center[0] - 0.1
y = mirror.center[1]
z = mirror.center[2]
set_position([x, y, z])
# done

# Query: "glass" to "table", hint: "onto"
```

```
glass = objects.movable
table = objects.goal
(max_x, max_y, max_z) = table.max_bounds
x = table.center[0]
y = table.center[1]
z = max_z + glass.size[2]/2
set_position([x, y, z])
# done

# Query: "phone" to "box", hint: "onto left"
phone = objects.movable
box = objects.goal
(max_x, max_y, max_z) = box.max_bounds
(min_x, min_y, min_z) = box.min_bounds
# Onto means that we need to choose the max_z
# Left on the box is the maximum in y-direction
x = box.center[0]
y = max_y
z = max_z + glass.size[2]/2
set_position([x, y, z])
# done

# Query: "stamp" to "table", hint: "lower left"
box = objects.movable
table = objects.goal
(min_x, min_y, min_z) = table.min_bounds
(max_x, max_y, max_z) = table.max_bounds
# The lower left corner of the tabletop is at maximum y and
# the minium x position
x = min_x
y = max_y
z = max_z + box.size[2]/2
set_position([x, y, z])
# done

# Query: "usb" to "cupboard", hint: "right"
usb = objects.movable
cupboard = objects.goal
(min_x, min_y, min_z) = cupboard.min_bounds
(max_x, max_y, max_z) = cupboard.max_bounds
# The right of the cupboard refers to the minimum y
x = cupboard.center[0]
y = min_y
z = max_z + usb.size[2]/2
set_position([x, y, z])
# done

# Query: "pen" to "paper", hint: "10cm right"
pen = objects.movable
paper = objects.goal
(min_x, min_y, min_z) = paper.min_bounds
# 10cm right of the paper refers to the minimum y of the paper
# minus 10cm
x = cupboard.center[0]
y = min_y - 0.1
z = max_z + usb.size[2]/2
set_position([x, y, z])
# done
```

```
# Query: "end effector" to "drawer", hint: "10cm push"
end_effector = objects.movable
drawer = objects.goal
normal = drawer.normal
# 10cm push means moving along the negative normal for 10cm
push_pos = end_effector.center - 0.1 * normal
set_position(push_pos)
# done
```

Constraint prompt:

```
from constraint_lib import (empty_constraint, full_constraint,
set_constraint)

# Query: "no constraint"
constraint = empty_constraint()
set_constraint(constraint)
# done

# Query: "exact"
# Exact means that everything is constrained
constraint = full_constraint()
set_constraint(constraint)
# done

# Query: "upright"
constraint = empty_constraint()
# Upright means there should be no rotation around the x or y axis
constraint.rotation_x = True
constraint.rotation_y = True
set_constraint(constraint)
# done

# Query: "horizontal plane"
constraint = empty_constraint()
# The x-y-plane is the horizontal plane
constraint.position_x = True
constraint.position_y = True
set_constraint(constraint)
# done

# Query: "z rotation"
constraint = empty_constraint()
constraint.rotation_z = True
set_constraint(constraint)
# done
```

