# OpenReview forum: "Language-guided Manipulator Motion Planning with Bounded Task Space"
_robot-learning.org/CoRL/2024/Conference — CoRL 2024_

### Official Review · Reviewer_1wvP · 2024-07-16
**Review of Language-Guided Manipulation Motion Planning with Bounded Task Space**

**Originality:** 1
**Technical Quality:** 2
**Clarity Of Presentation:** 2
**Potential Impact:** 1
**Recommendation:** 3
**Confidence:** 3

**Review:**

The presented planning method is validated in simulation and demonstrates modest improvements against a state of the art visual-language planning method. The modular framework provides an interesting perspective on using an LLM for discrete task planning while using an MPC-based controller for motion planning. While the method is interesting, there are several concerns.

Major Comments:
1. The presented framework is fundamentally a Task And Motion Planning (TAMP) method, yet there is no literature review on TAMP methods. In my opinion, this method does not use an LLM for motion planning, rather it uses the LLM for discrete task planning (move to, pick up, drop, etc), and so the lit review on language-based motion planning does not fit within this paper. Without this TAMP lit review, it is hard to evaluate the contributions of the paper. Furthermore, there is no discussion on other safety-based methods in the literature such as control barrier functions, reachability, etc.

2. The authors describe their method as “safe”, however safety is only enforced at discrete time-steps within the MPC pipeline. This does not guarantee the method will be safe while transitioning between discrete, optimal states. The authors may be interested in continuous-time safety methods such as [1] [2].

3. The novelty of the individual elements of the presented framework are not thoroughly discussed. It is unclear whether LLMs have been previously used for discrete task planning, however planning over convex sets using graph optimization (which the authors claim is a contribution) [3] and MPC-based obstacle avoidance have both been presented in the literature. While novelty may lie in the combination of these elements into a single planning framework, the experiments and results of this paper are not yet sufficiently rich enough to demonstrate this novelty.

4. The proposed method would be better validated with real-world hardware experiments.


[1] Jonathan Michaux, Adam Li, Qingyi Chen, Che Chen, Bohao Zhang, and Ram Vasudevan. "Safe Planning for Articulated Robots Using Reachability-based Obstacle Avoidance With Spheres." Robots: Science and Systems, 2024.

[2] Ian Mitchell. "Comparing forward and backward reachability as tools for safety analysis." In International Workshop on Hybrid Systems: Computation and Control, 2007.

[3] Jack Tobia, Pablo Parrilo, and Russ Tedrake. "Shortest paths in graphs of convex sets." SIAM Journal on Optimization, 2024.

[4] Russell Buchanan, Lorenz Wellhausen, Marko Bjelonic, Tirthankar Bandyopadhyay, Navinda Kottege, and Marco Hutter. "Perceptive whole‐body planning for multilegged robots in confined spaces." Journal of Field Robotics, 2021.

**Quality Of The Limitations Section:**

2

**Questions For Rebuttal:**

1. A more thorough literature review on safety-based planning methods as well as on task and motion planning (TAMP).

2. More simulation results are required in order to evaluate the novelty of the modular framework, as it is challenging to determine the novelty of each of the individual elements of the proposed method.

**Robotics Focus:**

2

**Summary Of Paper:**

The presented planning framework tackles task and motion planning by using an LLM for discrete task planning and for motion planning first building a set of of convex sets representing empty space, connects these sets via a graph, and plans over this graph.

**Summary Of Recommendation:**

It is hard to determine the novelty of the proposed modular task and motion planning framework without significantly more simulation and hardware experiments in more complex scenarios.

---

### Official Review · Reviewer_p2nF · 2024-07-21
**Language-guided Manipulator Motion Planning with Bounded Task Space**

**Originality:** 3
**Technical Quality:** 3
**Clarity Of Presentation:** 4
**Potential Impact:** 2
**Recommendation:** 3
**Confidence:** 3

**Review:**

Strengths
* The authors have identified an interesting area of research, namely zero-shot reasoning using open-vocabulary
* The authors demonstrate their method on hardware
* The authors’ method is outperforming VoxPoser

Weaknesses
* The novelty of the paper is unclear. There are many papers that use LLMs to generate motion plans for robots. In fact, there are methods that have more stringent safety constraints.
* The figures in the main part of the paper are very unclear. Arguably the most important part of Figure 1 is the image of the simulated robot performing the task. The paper could benefit from showing an image of the robot actually solving a task where the time steps of the arm are transparent and overlaid on the image.
* The authors emphasize safety, but only enforce safety of the end-effector. This seems very strange because there are numerous real-time trajectory optimization methods for manipulators that would have allowed the authors to enforce safety for the entire robot.
* The discussion on the novelty of their MPC method is rather limited?
* Why is the method called MPC? MPC usually refers to some kind of receding-horizon control. The authors description sounds like they are generating the plan offline.
* The comparisons in the paper are limited.
* The scenes in the paper are  rather simple.

**Quality Of The Limitations Section:**

3

**Questions For Rebuttal:**

* Why not show an example trajectory in the main overview figure?
* Can you motivate the use of GCS more?
* Can you discuss the novelty of your BoundMPC algorithm a bit more? It seems like you are using GCS, but only enforcing collision constraints for the end-effector. Is this because GCS isn’t actually real-time? Furthermore, I do not believe that the linear interpolation is novel. Isn’t this how TrajOpt works?
* Is it possible for you to do evaluate the performance of your method on more challenging scenes? How many obstacles can your method handle?
* How do you think your method would work for tasks requiring longer sequences?
* Could the authors explain the various pieces of their method a little better. They say BoundMPC runs at 10Hz and that their path planner takes 0.15s. Is GCS generating the path and BoundMPC tracking it? Is GCS running online or offline? BoundMPC is only generating torques at 10Hz? This sounds very slow for torque control.
* The grasp detection should be discussed more.

**Robotics Focus:**

4

**Summary Of Paper:**

The paper appears to be doing language-guided motion planning by combining LLMs and MPC.

**Summary Of Recommendation:**

I chose weak accept because I do not understand the novelty of their MPC method. But the combination of MPC with LLMs is still very compelling.

---

### Official Review · Reviewer_fPqJ · 2024-07-24

**Originality:** 2
**Technical Quality:** 3
**Clarity Of Presentation:** 3
**Potential Impact:** 2
**Recommendation:** 3
**Confidence:** 3

**Review:**

The paper is generally easy to follow. The components of the motion planning framework that are not part of this work (LLM, vision module, MPC controller) are mostly described with sufficient detail to understand the whole framework.

The main issue I have with the paper is a poor motivation of the path planner, the only component of the framework that is novel. The idea of constructing a piecewise linear reference path from a graph that connects overlapping convex sets of collision-free states is interesting, but the paper doesn't explain the need for this idea. The goal is to generate smooth trajectories, but the smoothness is achieved by the MPC controller, not by the path planner. In fact, the path planner outputs piecewise linear reference paths, which are not smooth. That being said, it is not clear to me why not any existing off-the-shelf path planner is used, and what the actual advantages of the proposed planner are. Since the main contribution of the paper is the path planner, the paper should discuss this in detail. Ideally, the paper should also provide an ablation by swapping the proposed planner with an existing one. For instance, what happens if VoxPoser's motion planner is used in conjunction with the MPC controller? I have the feeling that this would work just as well as using the proposed planner.

Some other comments:

- The paper uses "path planner", "motion planner" and "trajectory planner" interchangeably, which is confusing. Please be consistent to avoid confusion.
- Section 3.2.1 should be in the related work section. Please mention the used LLM instead.
- Line 182: \mathcal{D} is not defined.
- Line 182: What is matrix A_k and vector b_k? Please explain.

Post rebuttal: The authors have addressed my previous concerns regarding the motivation of the planner and the smoothness of trajectories.

**Quality Of The Limitations Section:**

3

**Questions For Rebuttal:**

- I'm surprised that a piecewise linear reference path is computed, since the main motivation of the planning and control modules are to generate smooth trajectories. Why not use B-splines or other curves for the reference path?
- Why does the framework have such a low success rate in the TeaCup problem?

**Robotics Focus:**

4

**Summary Of Paper:**

This paper proposes a modular motion planning framework, consisting of an LLM to generate Python code for the motion task completion, a vision module, a path planner to compute smooth paths of the robot, and a MPC controller to execute the path. For the LLM, vision module and MPC controller, existing approaches are used. The main novely lies in the path planner. It computes collision-free convex sets and connect them via a graph. The path for the robot to follow is then computed based on the shortest path in the graph. The motion planning framework is evaluated using simulated and physical manipulation tasks.

**Summary Of Recommendation:**

In summary, while the paper contains some interesting ideas, more discussion on the novelty of the path planner must be provided, including an ablation study with existing planners.

---

### Author Rebuttal · Authors · 2024-08-14

Dear Reviewers,

Please see the attached PDF file of the updated manuscript, the videos of the hardware and simulation experiments, and the reviewer response letters (equal to the official comments).

We hope that this rebuttal clears up all the valid criticism and makes you reconsider your publication recommendation.

Greetings

---

### Decision · Program_Chairs · 2024-09-04

**Decision:**

Accept

**Comment:**

This paper presents a planning framework that addresses task and motion planning (TAMP) by utilizing a Large Language Model (LLM) for discrete task planning and a graph of convex sets (GCS) for motion planning.

Before the rebuttal, the major concern among reviewers is the lack of novelty. The reviewers noted that while the components of the framework are existing work, the innovation may lie in their combination and significantly more simulation and hardware experiments in more complex scenarios are needed to justify the advancement enabled by this combination.

Other concerns include confusion regarding the terminologies and equations used, unclear figures, an emphasis on safety that only considers the end-effector, the lack of an ablation study with other existing planners, and an insufficient literature review on TAMP methods.

During the rebuttal, the authors did an excellent job addressing the reviewers’ concerns. As a result, all reviewers now rate the paper as a Weak Accept. While some reviewers still have concerns regarding the novelty of the paper, they acknowledge the clear motivation behind the work and the hardware experiments that demonstrate the proposed framework's effectiveness.